# Feeding efficiency of two coexisting nectarivorous bat species (Phyllostomidae: Glossophaginae) at flowers of two key-resource plants

Jan Philipp Bechler[1]*, Kira Steiner[1], Marco Tschapka[1,2]

**1** Institute of Evolutionary Ecology and Conservation Genomics, Ulm University, Ulm, Germany,
**2** Smithsonian Tropical Research Institute, Ancón, Panama City, Panama

* jan.bechler@alumni.uni-ulm.de

**Data Availability Statement:** All relevant data are within the manuscript and its Supporting information files.

## Abstract

Animals should maximize their energy uptake while reducing the costs for foraging. For flower-visitors these costs and benefits are rather straight forward as the energy uptake equals the caloric content of the consumed nectar while the costs equal the handling time at the flower. Due to their energetically demanding lifestyle, flower-visiting bats face particularly harsh energetic conditions and thus need to optimize their foraging behavior at the flowers of the different plant species they encounter within their habitat. In flight cage experiments we examined the nectar-drinking behavior (i.e. hovering duration, nectar uptake, and the resulting feeding efficiency) of the specialized nectar-feeding bat *Hylonycteris underwoodi* and the more generalistic *Glossophaga commissarisi* at flowers of two plant species that constitute important nectar resources in the Caribbean lowland rainforests of Costa Rica and compared nectar-drinking behavior between both bat species and at both plant species. We hypothesized that the 1) specialized bat should outperform the more generalistic species and that 2) bats should generally perform better at flowers of the nectar-rich flowers of the bromeliad *Werauhia gladioliflora* than at the relatively nectar-poor flowers of the Solanaceae *Merinthopodium neuranthum* that has an extremely long flowering phase and therefore is an extremely reliable nectar resource, particularly for the specialized *Hylonycteris*. While we did not find substantial differences in the feeding efficiency of the generalist *G. commissarisi*, we observed an increased feeding efficiency of the specialized *H. underwoodi* at flowers of the nectar-poor *M. neuranthum*. This suggests that familiarity and ecological importance are more important determinants of the interaction than just morphological traits. Our results demonstrate that in addition to morphology, behavioral adaptations are also important drivers that determine the fitness of nectar-feeding bats. Both familiarity with and the ecological importance of a resource seem to contribute to shaping the interactions between pollinating bats and their plants.

**Funding:** JPB was funded by the Landesgraduiertenförderung (LGF) (https://www.uni-ulm.de/einrichtungen/zuv/dez1/recht-und-organisation/stipendien/lgfg-stipendien/) and the Studienstiftung des Deutschen Volkes e.V. (https://www.studienstiftung.de). JPB and KS were both awarded a stipend by the Elisabeth-Kalko-Stiftung (https://www.regenwald-schuetzen.org/unsere-projekte/forschung-und-studien/forschungsstipendien). The funders had no role in study design, data collection and analysis, decision to publish, or preparation of the manuscript.

**Competing interests:** The authors have declared that no competing interests exist.

## Introduction

Optimal foraging theory predicts that animals should maximize during foraging net energy gain while minimizing energy expenditure [1, 2]. Balancing these costs and benefits is a crucial determinant for their ecological niche [3] and thus affects interspecific interactions as well as ecosystem functions. However, in natural systems, costs and benefits of foraging behavior are usually quite complex and often difficult to assess.

Mutualistic pollinator-plant interactions have been proven to be apt systems for the study of optimal foraging, as costs and benefits for both animal and plant partners are unusually transparent [2, 3]. Plants provide nectar for pollinators, which in turn mediate outcrossing by carrying pollen from one flower to another. Nectar secretion as an adaptive trait [3] demands that zoophilous plants maximize their fitness when they provide the lowest volumes of nectar that will attract just enough pollinators [4–7]. Moreover, flowers occur often rather dispersed, requiring the pollinator to cover costly distances between them. Hence, foraging efficiency, defined as the ratio between energy uptake and energy investment, is an important determinant of a pollinator's fitness.

Nectar uptake substantially influences the foraging success of pollinators in natural systems [8]. For larger pollinators that extract nectar while hovering in front of flowers, nectar uptake requires a particularly high energetic investment, as shown for hummingbirds [9–11], neotropical nectar-feedings bats [12, 13], sphingid moths [14], and to some extent sunbirds [15]. Hence, a high efficiency in the extraction of nectar is key to increase net caloric yield, especially for larger flower-visitors and consequently for the persistence of the interaction [3, 16–19]. Therefore, a comprehensive study of these parameters, including nectar uptake, hovering duration and their interplay that together determine feeding efficiency of a species is imperative for a better understanding of resource choice and resource partitioning within a community.

Thus far, important baselines on these nectar-feeding parameters have been gained from experiments with artificial feeders under *ad-libitum* conditions [e.g. 20–33]. However, the feeding behavior of pollinators at real flowers may differ substantially from their behavior under experimental conditions [3]. The combination of observational and experimental data allows gaining a deeper understanding of foraging decisions [34]. Recent studies demonstrated that floral morphology may influence pollinators' feeding behavior [20, 21, 35] as these structures have the potential to facilitate or hinder access to nectar rewards [20, 22, 36], and thus may lead to specific preferences for certain plant species [3]. Data on the foraging efficiency of most pollinators under natural conditions remain rather elusive as the microscopic scale of most of these interactions impedes accurate quantification. In contrast, systems involving relatively large vertebrate pollinators may allow for the acquisition of otherwise inaccessible information.

Actually, nectar-feeding bats may serve as a great model for the investigation of foraging efficiency. Being large, flying, endothermic pollinators feeding on nectar from often widely dispersed flowers, bats face particularly harsh energetic conditions and thus are under strong selective pressure for efficient foraging [37–39]. Consequently, chiropterophilous flowers produce among all pollination syndromes the highest nectar volumes for attracting their large pollinators [4, 38, 40]. The macroscopic scale of this system allows the quantification of different parameters of the drinking behavior of bats under realistic conditions.

*Hylonycteris underwoodi* and *Glossophaga commissarisi* (Phyllostomidae: Glossophaginae) comprise the year-round resident species of the nectar-feeding bat guild in the Caribbean lowland rainforests of Costa Rica [41]. These two species differ morphologically and in their degree of specialization to a nectar diet. The 10 g *G. commissarisi* (Fig 1A) is a relatively generalistic flower-visiting bat that regularly supplements its diet with fruits and insect [41–44].

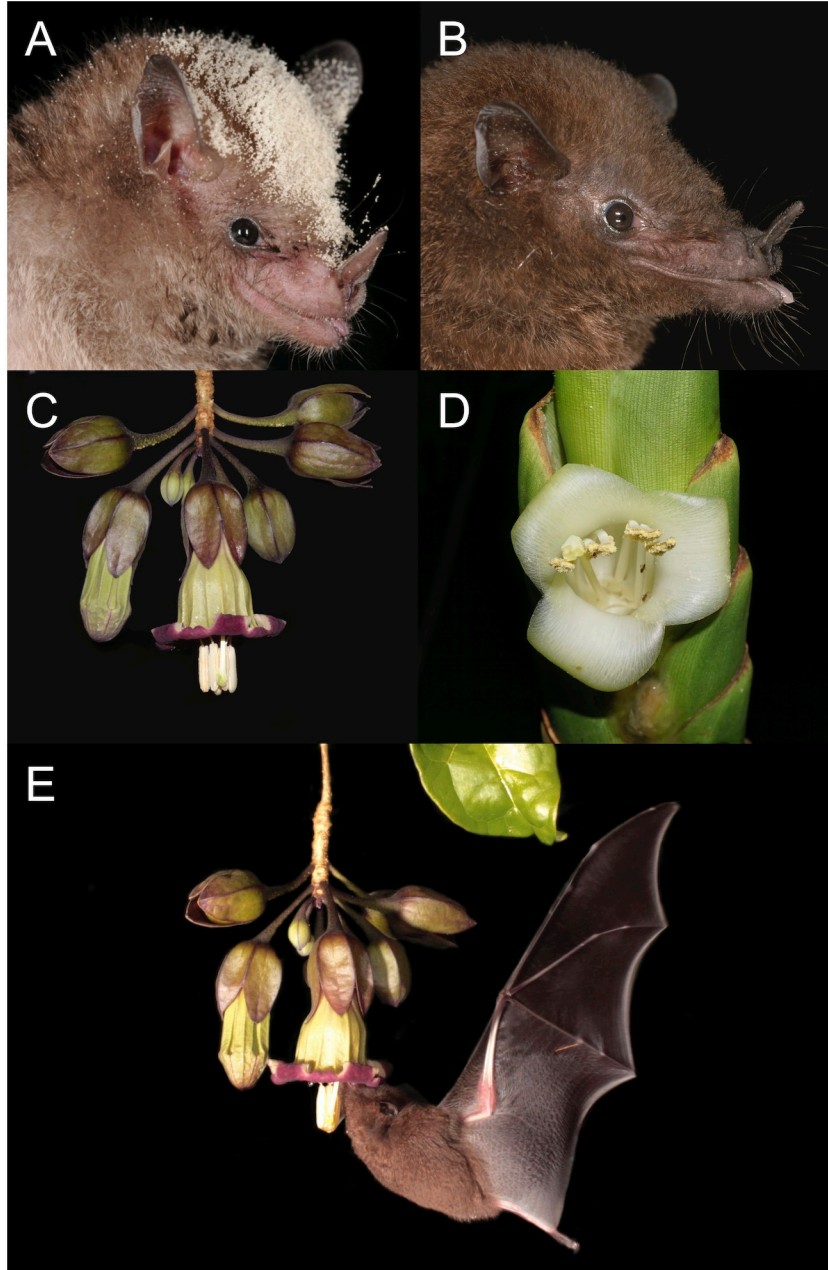

**Fig 1. The resident nectar-feeding bat guild and two key nectar resources.** The two resident nectar-feeding bat species, *Glossophaga commissarisi* (A) and *Hylonycteris underwoodi* (B), and the pendulous flower of *Merinthopodium neuranthum* (C) and the horizontally oriented *Werauhia gladioliflora* (D) at La Selva Biological Station, Costa Rica. When visiting a flower of *M. neuranthum*, bats (here *H. underwoodi*) raise their head to the flower above and lap the nectar (E).

Overall, *G. commissarisi* preferably uses nectar, but switches seasonally to feeding on fruits during periods of nectar-shortage [42, 45]. In contrast, the more specialized 8 g *H. underwoodi* (Fig 1B) depends all year round highly on floral nectar as their morphological adaptations, specifically the skull shape, impedes them from efficiently using alternative food resources that require a higher bite force [25, 46, 47].

Over the course of a year, the species select their floral resources mainly based on the respective differences in phenology, nectar quantity and flower density [41]. Among the locally available plants, *Merinthopodium neuranthum* (Solanaceae) and *Werauhia gladioliflora* (Bromeliaceae) represent two of the key nectar resources in the area [41, 48, 49]. The hemiepiphyte *M. neuranthum* only occurs at low densities throughout the forest, but due to a 10–11 month flowering period it offers a highly predictable resource that is consistently used by the bats, especially by the highly nectar-dependent *H. underwoodi*. During this time plants produce very reliably almost daily a low number of pendulous, bell-shaped flowers (Fig 1C) that secrete relatively small amounts of nectar during a short anthesis from sunset until ca. 0100 am [41, 48]. In contrast, *W. gladioliflora* occurs in large population densities in more open areas, such as forest clearings and along river edges. Individuals open during the exclusively nocturnal anthesis mostly a single, funnel-shaped, horizontally oriented flower (Fig 1D) with relatively large nectar volume [49] over a few weeks in October and November. Although just temporarily, they provide due to their locally high population densities a high level of nectar availability [41].

The nectar-feeding bat community of La Selva Biological Station in the Caribbean lowland rainforest of Costa Rica has been studied to great detail [41, 42, 49, 44, 45, 50], especially in association with the changing resource availability over the annual cycle. This interesting system, featuring a generalist and a specialist resident nectar-feeding bat species in concert with extensive information on the available nectar resources makes La Selva Biological Station an excellent location for studies on the feeding behavior of nectar-feeding bats at real flowers. In this study we aim to compare the drinking efficiency of the generalist *G. commissarisi* and the specialist *H. underwoodi* at flowers of the two important, but distinctly different nectar resources *M. neuranthum* and *W. gladioliflora*.

We hypothesize that the degree of morphological specialization in nectarivorous bats influences their feeding efficiency, with more specialized species demonstrating higher feeding efficiency compared to generalist species. Additionally, we hypothesize that the nectar volume provided by flowers affects the feeding efficiency of bats, with larger volumes facilitating a higher nectar uptake rate. From these hypotheses, we derive the following specific predictions:

1. We predict that due to its higher degree of morphological specialization, *H. underwoodi* generally achieves a higher feeding efficiency than the more generalist *G. commissarisi*.

2. Further, we predict that both bat species feed more efficiently at the nectar-rich flowers of *W. gladioliflora* than at the relatively nectar-poor flowers of *M. neuranthum*.

## Material and methods

### Field site and bat capture

Fieldwork was conducted at La Selva Biological Station in the Caribbean lowland rainforest of Costa Rica (10˚ 25’ 19.2” N, 84˚ 0’ 54” W) between September 2017 and December 2019. Bats were captured with mist nets in close proximity to flowering plants. We only used adult, non-pregnant and non-lactating individuals for our experiments. Body mass was recorded with a Pesola spring balance (± 0.5 g) and forearm length was measured with a caliper (± 0.01 mm). The experiments were conducted with single individuals at a time. The bat was released into a flight tent (EUREKA, 4.5 m x 4.5 m), set up at a shady part of the La Selva laboratory clearing. During the first night (i.e., night of bat capture), an artificial feeder with *ad-libitum* honey-water was offered and the bat was allowed to habituate to the new situation, while the

experiments started only in the second night. After the experiments, the bat was released at the capture site. We strictly followed the guidelines recommended by the American Society of Mammalogists [51].

## Experimental setup

For the experiments on feeding efficiency we prepared a honey-water solution with a sugar concentration of 16%, which was measured with a handheld refractometer (Krüss Co., Hamburg, Germany; range 0–30% weight/weight) and corresponds to the naturally occurring mean sugar concentration of chiropterophilous flowers [52]. Density of this solution was empirically determined to be 1.09 g × ml$^{-1}$. Although the composition of sugars in floral nectar varies between species, nectar of chiropterophilous plants commonly consists primarily of glucose and fructose [53–55]. Similarly, honey consists primarily of about equal parts of the hexoses fructose and glucose [56]. In previous studies, the use of honey-water proved to be an apt approach, as honey emits a slight odor, which facilitates for the bats the localization of a food source in a novel environment (personal observation JPB, MT). For the experiments we used freshly opened flowers from the two key resource plants *M. neuranthum* and *W. gladioliflora.* Though varying in nectar sugar concentration over the course of the anthesis, the 16% nectar concentration used in the experiments falls well within the range of the observed nectar concentration in both plant species [43, personal observation JPB, MT]. At the beginning of an experiment, already secreted nectar was removed with a glass capillary tube (100 μl, ± 0.25%, 20°C) attached to a syringe, so we had full control over the nectar in the flower. Floral depth and diameter was measured using a caliper (± 0.01 mm). We aimed to quantify the parameters nectar uptake [ml] and hovering duration [s] of bats at natural flowers under realistic conditions. Nectar uptake represents the benefit of a flower visit, while hovering duration stands for the energetic cost of the visit. The quotient of these parameters therefore represents a benefit-to-cost ratio, the feeding efficiency [ml × s$^{-1}$] [23, 27].

We attached the flowers to an analytical laboratory balance (Mettler Labstyle 152 and Kern PCB 250–3; precision 1 mg) in a manner closely resembling its natural presentation on the plant. Flowers of *W. gladioliflora* were presented horizontally in an artificial inflorescence that was rebuilt out of hard foam and fixed to the weighing plate (Fig 2A). For experiments with the pendulous flowers of *M. neuranthum*, the scale was placed on an acrylic glass plate suspended from the tent's roof. A hole in the glass plate allowed suspension of the flowers from the under-floor weighing appliance (Fig 2B). Using a reflection light sensor (Datalogic S100-PR-2-A00-PK) and a reflector we placed an infrared light beam in front of the flower openings, which allowed recording the duration of each bat visit during the experiments. Prior to the start of the experiment, bats were permitted to visit the flower filled with honey-water several times to ensure successful habituation to the new situation. For the experiments the balance was set to zero and flowers were filled in a random order with a various quantities of honey-water, covering the naturally possible range of nectar volumes, i.e. from 5 mg to 250 mg in flowers of *M. neuranthum* and from 5 mg to 1,000 mg in flowers of *W. gladioliflora.*

We used the custom-built 'Covfefe-system' (computer-based verification of feeding efficiency in field experiments) to acquire data on the different parameters of the bats' nectar-feeding behavior during the experiments. A program written in LabVIEW14® (National Instruments) logged the time of each status change of the sensor with a precision of 1 ms into a text-file. Simultaneously, the weight of the flower was recorded every second and also written into a text file. An R script was used to subsequently calculate the time and duration of the flower visit and match it with the corresponding change in weight. Bat visits were then recorded until the flower was depleted. To account for potential weight loss of the flower

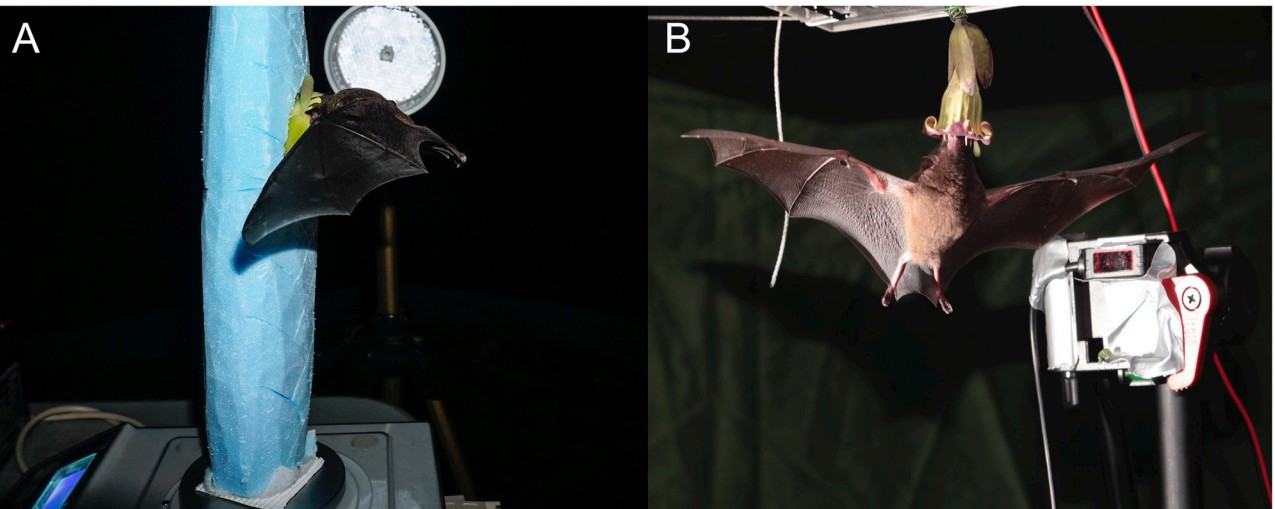

**Fig 2. Experimental setup in the flight cage.** *H. underwoodi* drinking from the horizontally oriented flower of *W. gladioliflora* presented in an artificial inflorescence (A) and from the downward-hanging flower of *M. neuranthum* (B). Both the IR-light-sensor (B) and the reflector (A) are visible in the background.

through desiccation, the balance was always set to zero before a new amount of honey-water was offered to the bat.

For further analysis, the weight of honey-water provided in the flowers was converted into the encountered volume $V_{en}$ [ml], based on the honey-water density of 1.09 g × ml$^{-1}$. The mass of the extracted honey-water was converted into the extracted volume $V_{ex}$ [ml] analogously. We calculated the standardized nectar extraction efficiency ($E_s$) [ml × s$^{-1}$], by including the estimated daily energetic expenditure of each individual bat (*DEE*) [kJ × day$^{-1}$] [23], thus allowing to account for individual size variation.

## Statistical analysis

Data analysis was carried out in R v. 3.6.1 [57]. Data exploration was adapted following Zuur et al. [58]. Some flower visits were characterized by exceptionally short duration (faulty alignment of the infrared-light sensor) or extremely small amounts of consumed honey-water (no real drinking events). We used Mahalanobis distances, a bivariate approach, to detect and remove such outliers. The variables (duration and $V_{ex}$) were log-transformed to increase sensitivity for the lower range of each variable. The threshold was set to the product of the mean of Mahalanobis distances and the extremeness degree k (k = 1.5 × STD$_{MD}$). Of a total of 1,965 registered visits at real flowers, 45 visits were classified as outlier and removed from the dataset.

We used linear mixed effect models from the *lme4* package [59] to compare hovering duration, nectar uptake ($V_{ex}$), and standardized nectar extraction efficiency ($E_s$) between bats and plant species. Fixed effects were '$V_{en}$' × 'bat species' and '$V_{en}$' × 'plant species', respectively. Dependent variables were 'hovering duration', '$V_{ex}$', and '$E_s$'. To meet the assumptions for linear mixed models we square root—transformed the $V_{en}$ and the dependent variables. Individual bats were included as random effect.

## Ethics statement

The study was approved by the *Ministerio de Ambiente y Energía de Costa Rica* (MINAE). Research permits were granted by the MINAE (Resolutions: 054-2017-ACC- PI,

SINAC-ACC-PI-R-036-2018, SINAC-ACC-PI-R-054-2019, SINAC-ACC-PI-R-112- 2019, SINAC-ACC-PI-R-105-2019).

## Inclusivity in global research

Additional information regarding the ethical, cultural, and scientific considerations specific to inclusivity in global research is included in the Supporting Information (S1 Checklist).

## Results

### General observation

After initial habituation, bats behaved uniformly by regularly drinking from the flowers in bouts of several visits in rapid succession, followed by longer resting breaks (ca. 30–60 minutes). We tested a total number of 11 *H. underwoodi* and 8 *G. commissarisi* individuals, resulting in a total number of 1,920 flower visits. At flowers of *M. neuranthum*, we recorded 383 visits by *H. underwoodi* (n = 9 individuals) and 272 by *G. commissarisi* (n = 5 individuals). At *W. gladioliflora*, we recorded 612 visits by *H. underwoodi* (n = 5 individuals) and 653 visits by *G. commissarisi* (n = 6 individuals). In the beginning of the experiments, all individuals hovered in front of the flowers. However, a few individuals started to cling to flowers of *M. neuranthum*. We observed 8 clinging events (2.0% of total flower visits) by *H. underwoodi* (n = 2 individuals) and 60 (18.1% of total flower visits) by *G. commissarisi* (n = 5). We excluded those visits from the further analysis, as prior field video recordings revealed that clinging of bats at flowers of *M. neuranthum* occured only very rarely under natural conditions.

### Nectar-drinking behavior of glossophagine bats at real flowers

**Hovering duration.**   The hovering duration represents the immediate energetic investment of a bat when drinking nectar from a flower. At flowers of *M. neuranthum* we found no significant difference in hovering duration between the two bat species and no significant effect of the interaction between bat species and the encountered nectar volume $V_{en}$ within the flower (LMM: bat species: $F_{1, 260.5} = 0.198$, p = 0.657; $V_{en}$: $F_{1, 642.5}$, p < 0.0001; bat species × $V_{en}$: $F_{1, 642.5} = 0.191$, p = 0.662; conditional $R^2 = 0.603$). However, when bats visited flowers of *W. gladioliflora*, we found a significant effect of the interaction between bat species and $V_{en}$ (LMM: bat species: $F_{1, 9.4} = 0.028$, p = 0.870; $V_{en}$: $F_{1, 1,257.6} = 141.242$, p < 0.0001; bat species × $V_{en} = F_{1, 1,256.6} = 44.486$, p < 0.0001; conditional $R^2 = 0.607$), with *G. commissarisi* increasing its visit duration at a higher rate with increasing $V_{en}$ than *H. underwoodi* (Fig 3A and 3B).

**Extracted volume ($V_{ex}$).**   The extracted volume $V_{ex}$ represents the caloric uptake a bat accomplished during the flower visit and thus is crucial in determining the profitability of the visit. With increasing $V_{en}$, $V_{ex}$ increased at flowers of *M. neuranthum* at a significantly higher rate for *H. underwoodi* than for *G. commissarisi* (LMM: bat species: $F_{1, 22.9} = 0.047$, p = 0.830; $V_{en}$: $F_{1, 646.1} = 1,705.741$, p < 0.0001; bat species × $V_{en}$: $F_{1, 646.1} = 10.045$, p = 0.002; conditional $R^2 = 0.767$). In turn, at flowers of *W. gladioliflora* $V_{ex}$ increased at a higher rate for *G. commissarisi* than for *H. underwoodi* (LMM: bat species: $F_{1, 9.8} = 0.016$, p = 0.902; $V_{en}$: F1, 1,260.4 = 1,979.233, p < 0.0001; bat species × $V_{en}$: $F_{1, 1,260.4} = 95.619$, p < 0.0001; conditional $R^2 = 0.734$). *Glossophaga commissarisi* extracted from flowers of *W. gladioliflora* overall significantly higher volumes than *H. underwoodi* (*post hoc* test: p = 0.04) (Fig 3C and 3D).

**Standardized nectar extraction efficiency ($E_s$).**   At flowers of *M. neuranthum* both species showed a significant increase in $E_s$ with increasing $V_{en}$, however, no significant effect of bat species or their interaction with $V_{en}$ was found (LMM: bat species: $F_{1,30.5} = 0.094$, p = 0.762; $V_{en}$: $F_{1, 646.8} = 1,0534.000$, p < 0.0001; bat species × $V_{en}$: $F_{1, 646.8} = 0.009$, p < 0.0001;

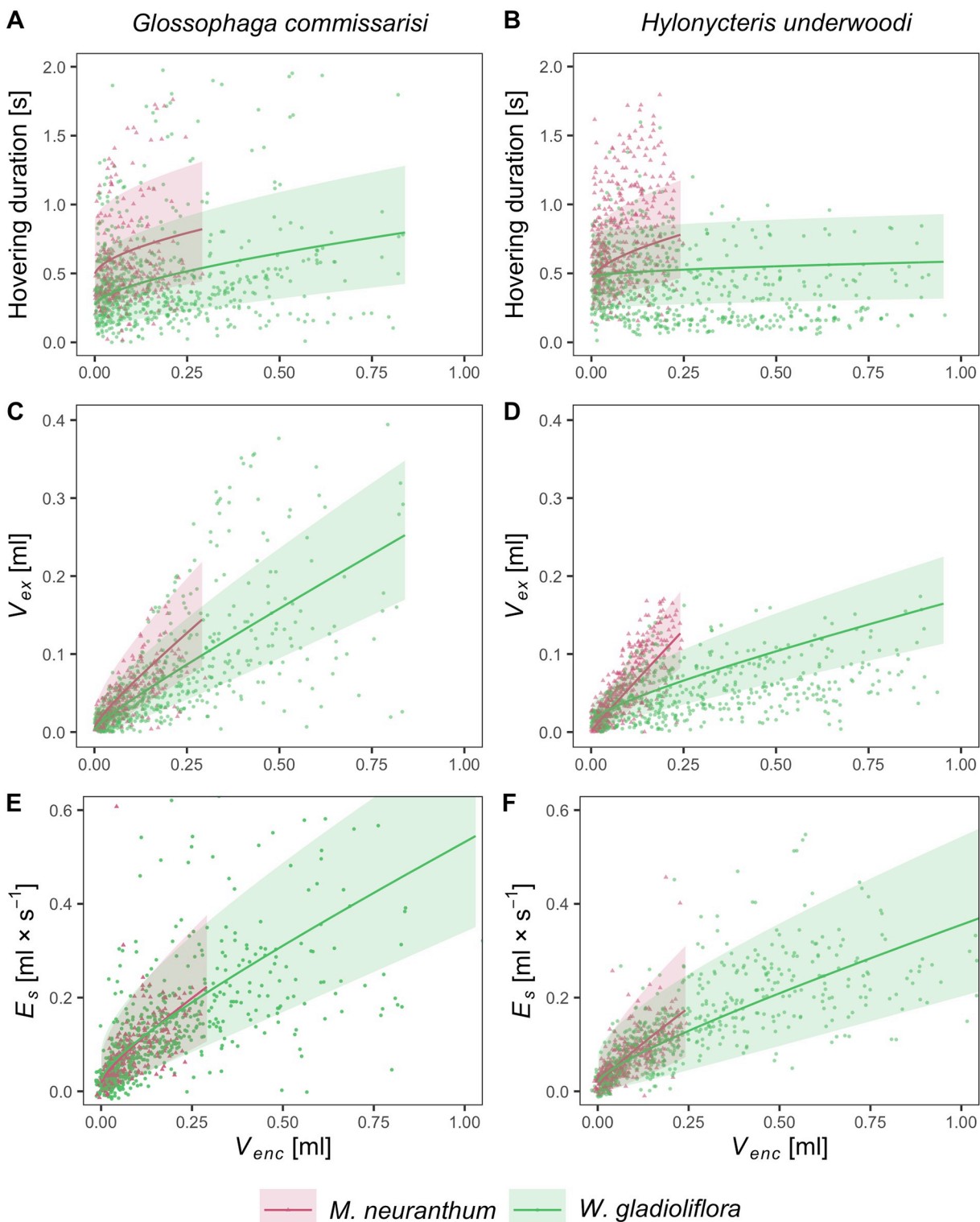

**Fig 3. Comparison of nectar-drinking behavior of the two nectar-feeding bat species at the two resource plants.** (A, B) Hovering duration, (C, D) extracted nectar volume, (E, F) standardized nectar extractions efficiency. 9 individuals of *H. underwoodi* performed 383 drinking events at flowers of *M. neuranthum* and 5 individuals accomplished 612 drinking events at flowers of *W. gladioliflora*. For *Glossophaga commissarisi*, we observed 5 individuals with 272 drinking events at flowers of *M. neuranthum* and 6 individuals with 653 drinking events at flowers of *W. gladioliflora*.

conditional $R^2$ = 0.670). In contrast, the interaction between $V_{en}$ and bat species had a significant effect on extraction efficiency at flowers of *W. gladioliflora* (LMM: bat species: $F_{1, 9.47}$ = 0.1135, p = 0.7436; $V_{en}$: $F_{1, 1,259.37}$ = 1,2276.5179, p < 0.0001; bat species × $V_{en}$: $F_{1, 1,259.37}$ = 0.0086, p < 0.001; conditional $R^2$ = 0.692), with *G. commissarisi* increasing its extraction efficiency at a higher rate with increasing $V_{en}$ than *H. underwoodi* (Fig 3E and 3F).

## Differences between the two bat species

**Hovering duration.** Both bat species significantly extended their hovering duration with increasing $V_{en}$ at flowers of both plant species. For *G. commissarisi*, plant species also had a significant effect on hovering duration, but not the interaction between plant species and $V_{en}$ (LMM: plant species: $F_{1, 917.1}$ = 34.068, p < 0.0001; $V_{en}$: $F_{1, 915.3}$ = 64.116, p < 0.0001; plant species × $V_{en}$: $F_{1, 915.2}$ = 0.062, p = 0.803; conditional $R^2$ = 0.609). On average, *G. commissarisi* hovered 0.176 s longer at flowers of *M. neuranthum* than at flowers of *W. gladioliflora* (*post hoc* test: p < 0.0001), but increased hovering duration at both flowers at the same rate with increasing $V_{en}$ (Fig 3A). For *H. underwoodi* we found no significant effect of plant species on hovering duration, but of the interaction between plant species and $V_{en}$ (LMM: plant species: $F_{1, 990.4}$ = 0.857, p = 0.355; $V_{en}$: $F_{1, 983.7}$ = 49.784, p < 0.0001; species × encountered: $F_{1, 983.6}$ = 24.840, p < 0.0001; conditional $R^2$ = 0.579). Whereas hovering duration at *W. gladioliflora* increased only slightly, hovering duration lasted consistently longer at *M. neuranthum* (*post hoc* test: p < 0.001) and increased significantly faster with increasing $V_{en}$ than at flowers of *W. gladioliflora* (Fig 3B).

**Extracted volume ($V_{ex}$).** For both species, $V_{ex}$ increased significantly with increased encountered nectar volume. For *G. commissarisi*, $V_{en}$ was the only significant effect on the extracted volume (LMM: plant species: $F_{1, 917.6}$ = 2.478, p = 0.1158; $V_{en}$: $F_{1, 917.8}$ = 1,086.537, p < 0.0001; plant species × $V_{en}$: $F_{1, 917.5}$ = 0.721, p = 0.396; conditional $R^2$ = 0.734) (Fig 3C). However, for *H. underwoodi* plant species had also a significant effect on $V_{ex}$, but not the interaction between $V_{en}$ and plant species (LMM: plant species: $F_{1, 988.6}$ = 65.777, p < 0.0001; $V_{en}$: $F_{1, 986.0}$ = 147.915, p < 0.0001; plant species × $V_{en}$: $F_{1, 985.8}$ = 219.332, p = 0.431; conditional $R^2$ = 0.743). Hereby, *H. underwoodi* consistently extracted a higher volume from a flower of *M. neuranthum* than from *W. gladioliflora* (*post hoc* test: p < 0.0001). The extraction increased significantly faster with increasing $V_{en}$ at flowers of *M. neuranthum* than at *W. gladioliflora* (Fig 3D).

**Standardized nectar extraction efficiency ($E_s$).** Both species showed a significant increase in nectar extraction efficiency with increasing $V_{en}$. Neither the plant species, nor the interaction between plant species and encountered nectar volume had a significant effect on $E_s$ in *G. commissarisi* (LMM: plant species: $F_{1, 917.8}$ = 2.421, p = 0.120; $V_{en}$: $F_{1, 915.6}$ = 1,975, p < 0.0001; plant species × $V_{en}$: $F_{1, 915.51}$ = 1.711, p = 0.191; conditional $R^2$ = 0.686) (Fig 3E). However, for *H. underwoodi*, plant species and their interaction with encountered nectar volume had a significant effect on $E_s$ (LMM: plant species: $F_{1, 866.5}$ = 11.068, p < 0.0001; $V_{en}$: $F_{1, 991.0}$ = 1,016.843, p < 0.0001; plant species × $V_{en}$: $F_{1, 991.0}$ = 32.782, p < 0.0001; conditional $R^2$ = 0.713), with *H. underwoodi* foraging at significantly higher $E_s$ at flowers of *M. neuranthum* than at *W. gladioliflora* (*post hoc* test: p < 0.0001) (Fig 3F).

## Discussion

### Differences between plant species

We could not confirm our hypothesis that glossophagine bats generally show a higher feeding efficiency at flowers of *W. gladioliflora*. Contrary to our assumption, *H. underwoodi* fed more efficiently at flowers of *M. neuranthum*, while *G. commissarisi* did not show a difference in

efficiency between the two offered flowers. Both bat species, however, had longer handling times and extracted also larger nectar volumes at flowers of *M. neuranthum* than at the flowers of *W. gladioliflora*.

The longer handling times at the pendent flower of *M. neuranthum* is not surprising as bats usually approach the horizontally oriented flowers of *W. gladioliflora* in a short, almost straight horizontal maneuvering flight (S1 File) and may even support their body in flight slightly on the sturdy inflorescence. In contrast, the approach to the dangling flowers of *M. neuranthum* is more complex. Bats have to maneuver more gently and generate sufficient uplift against gravity for inserting their head into the flower. This more complex task consequently results in longer visits at flowers of *M. neuranthum* (S2 File). As the upwards maneuvering is accomplished in hovering flight, the energetically most expensive mode of flight [10, 12, 13, 60], this phase represents a relevant energetic cost of flower exploitation. Floral traits are an important factor shaping flower visits. Corolla curvature [e.g. 22, 24, 34, 61], opening width [e.g. 25, 26, 62, 63], and floral tube length [e.g. 3, 16, 20–23, 27, 34, 39, 63–65] all have an impact on pollinators nectar-drinking behavior. Corolla curvature, depth, and width are rather similar in *W. gladioliflora* and *M. neuranthum*. The most striking difference, however, is the orientation of the flowers. Only few studies have addressed the role of flower orientation on pollinators' feeding behavior [11, 28, 66–69], and to our best knowledge this trait has not been examined in nectar-feeding bats thus far.

Similar to our findings, increased handling time at downwards-facing flowers was shown for hummingbirds feeding on *Penstemon* flowers [67] and for Australian honeyeaters [22]. However, Montgomerie [29] and Collins [22] could not confirm this finding for other hovering hummingbirds, and neither Ngcamphalala et al. [21] for the perching South African sunbirds in studies on the drinking behavior at artificial flowers. Nevertheless, independent of handling time, hummingbirds show higher energetic investment at pendulous flowers as their metabolic rate increased by up to 10% when feeding on flowers of that type [11]. Interestingly, hummingbird-pollinated plant species comprise many downward-facing flowers [66, 70, 71]. It has been argued that this trait has evolved as a deterrent for non-pollinating insects or to prevent rain to dilute the nectar [66], but it might also increase the pollinator's exploitation cost, thus illustrating the constant trade-off between costs and benefits for both partners.

Glossophagine bats have brush-like, hemodynamic papillae at the tip of their tongue and when their tongue is submerged in nectar, erection of papillae results in an increase in their nectar uptake ability [72]. In the downward-facing flowers of *M. neuranthum*, gravity might further enhance nectar uptake by enhancing nectar flow onto the tongue, compared to the horizontally oriented flowers of *W. gladioliflora*. Our results support the assumption by Kingsolver and Daniels [30] that gravity might increase the nectar uptake at pendulous flowers. While experiments with artificial flowers could not confirm this assumption [22], data from experiments with real flowers are missing thus far. We suggest that the consistent differences in drinking behavior of our nectar-feeding bats were mainly due to flower orientation, and we encourage further research on the role of this factor on pollinators' nectar-drinking behavior.

## Differences between bat species

We could not confirm our hypothesis that the specialized *H. underwoodi* generally feeds more efficiently than the generalist *G. commissarisi*. Although *H. underwoodi* was able to extract larger nectar volumes from *M. neuranthum* than *G. commissarisi* while not showing longer handling times, *H. underwoodi* did not show a significantly increased efficiency at *M. neuranthum*. Interestingly, *H. underwoodi* both extracted smaller nectar volumes and invested less time in the exploitation of flowers of *W. gladioliflora* compared to *G. commissarisi*.

Despite its smaller size, *H. underwoodi* was able to extract larger nectar volumes from flowers of *M. neuranthum* than *G. commissarisi*. Depending on the opening size, *H. underwoodi* is able to reach with its longer tongue between 15% and 40% deeper into artificial flowers than *G. commissarisi* [26]. Consequently, when presented with equal nectar volumes, *H. underwoodi* has a larger contact zone between nectar and tongue and can also extract smaller volumes more readily. Supporting our hypothesis, this advantage gets particularly obvious at smaller encountered nectar volumes, where *H. underwoodi* generally extracted higher portions than *G. commissarisi* and depleted flowers of *M. neuranthum* more completely than flowers of *W. gladioliflora*.

Intriguingly, the two bat species showed different patterns in nectar uptake efficiency. *Glossophaga commissarisi* increased its feeding efficiency with increasing nectar volume at both flowers at the same rate, thus showing no difference in efficiency at both flowers, while feeding efficiency in *H. underwoodi* increased when drinking at *M. neuranthum* at a higher rate, indicating a disproportionally increased nectar uptake with increased handling time. Therefore, the larger contact zone between tongue and nectar facilitated by a longer operational tongue length does not solely suffice to explain this difference in extraction efficiency in *H. underwoodi*.

The generalist *G. commissarisi* exhibits a high dietary flexibility by supplementing its diet with insects and fruits, and even seasonally switching to a predominantly frugivorous diet when floral resources are sparse [41–44]. In contrast, *H. underwoodi* shows a higher dependence on nectarivory, with cranial adaptations such as a more reduced dentition, a longer tongue and the associated longer rostrum better supporting the extraction of nectar from flowers [73]. Despite the extremely rare observations of seeds in fecal samples [41, 74], frugivory in *H. underwoodi* is presumably a mere emergency behavior when nectar availability is very low. Following a period of staggered flowering chiropterophilous plants maintaining a high nectar density at La Selva from October until February, *M. neuranthum* is during the subsequent time between March and September the predominant resource of nectar for glossophagine bats [41]. Besides its cranial adaptation on nectar-feeding, a particular wing morphology might *H. underwoodi* allow to move rather fast between dispersed flowers such as *M. neuranthum* [41]. Due to the high reliability of *M. neuranthum* in concert with its low population density, *H. underwoodi* might exploit *M. neuranthum* using so called traplines. This means, that they could forage over the night along roughly circular routes [75] with repeated visits to the same flowers timed to nectar production rate, and therefore reduce energetic investment in inspecting flights. This might consequently decrease the overall foraging cost so that *H. underwoodi* can very efficiently exploit *M. neuranthum*, despite its low nectar content and a low flower density [41].

The high predictability of *M. neuranthum* as nectar resource makes the Solanaceae a key resource for the nectar-dependent *H. underwoodi*, with particular importance during the time of low flower availability [41]. The resulting high familiarity of *Hylonycteris underwoodi* with *M. neuranthum* might be an important factor explaining the high feeding efficiency at these flowers.

The optimized exploitation of their key resource plant entails an overall reduction in energetic cost at a maximized nectar uptake. Klumpers et al. [3] suggested that pollinators should prefer flowers that allow the highest feeding efficiency. For our system this implicates that even when flowers of *W. gladioliflora* are abundant, *M. neuranthum* flowers might -due to the high familiarity—be still very profitable for *H. underwoodi*, despite the larger distances among the flowers. Competition is also often a relevant factor in shaping a plant-pollinator interaction [34, 76–79], which plays a particular role in bat-pollination systems. Despite the morphological differences between nectar-feeding bat species, in general all species can access all

chiropterophilous flowers present in a habitat. The larger *G. commissarisi* is about seven times more abundant in the area than *H. underwoodi* [41]. Additionally, during the season of flower abundance the flower-visiting species *Lichonycteris obscura* (Phyllostomidae: Glossophaginae) and *Lonchophylla robusta* (Phyllostomidae: Lonchophyllinae) are also coming into the area [41]. Those seasonal visitors likely feed predominantly on the mass flowering high-quality *W. gladioliflora* but not on the comparably rarer flowers of *M. neuranthum*. Thus, staying with the less profitable, yet more familiar flowers of *M. neuranthum*, rather than facing competition at the high quality resource *W. gladioliflora*, represents a relatively safe foraging strategy for *H. underwoodi*, making the species regionally more a *Merinthopodium* specialist than just a flower specialist.

## Conclusion

Foraging efficiency is an important driver in the evolution of plant-pollinator interactions that shapes preferences and the ability to utilize different resources. Many studies have so far focused on morphological matching between flowers and their pollinators and its impact on feeding efficiency. This is particularly true for insect-pollination systems, in which morphology might completely impede a flower visit or alternatively allow only certain species to monopolize a certain flower. Bat pollination, in contrast, is rather generalistic as the pollinator's morphology is not the main determinant that shapes the interaction. Although bats exhibit certain morphological adaptations to their foraging niche, their hardware (i.e. morphology) might be–at least in similar-sized species—less key to their ecological success than their software (i.e. behavioral complexity), like it is typically observed in mammals. We suggest that both the familiarity with and the ecological importance of a resource over the entire annual cycle play additional, yet crucial roles in shaping the interaction between pollinating bats and plants. Further, we encourage future experiments with real flowers.

## Supporting information

**S1 Checklist. Questionnaire on inclusivity in global research.**
(PDF)

**S1 Data.**
(XLSX)

**S1 File. *Hylonycteris underwoodi* drinking from a flower of *Werauhia gladioliflora*.** Slow-motion video (240 fps) of a *H. underwoodi* visiting a flower of *W. gladioliflora*. Inserting its head into the horizontally oriented flowers occurs with more ease compared to visiting a flower of *M. neuranthum*.
(MOV)

**S2 File. *Hylonycteris underwoodi* drinking from a flower of *Merinthopodium neuranthum*.** Slow-motion video (240 fps) of a *H. underwoodi* visiting a flower of *M. neuranthum*. Maneuvering its head upwards into the pendulous flower is more tedious compared to drinking from the horizontally oriented flower of W. gladioliflora.
(MOV)

## Acknowledgments

We thank the wonderful crew at La Selva Biological Station for their invaluable help during our fieldwork. Specials thanks go to Orlando Vargas, Bernal Matarrita, Enrique Castro, Danilo Brenes, Marisol Luna, and Greivin Salazar. Muchas gracias! Many thanks also go to Diego

Dierick for his enormous patience and for always sharing his technical genius. Further we thank Monika Springer and Bernal Rodríguez Herrera from the Universidad de Costa Rica for their friendly support during our stays in Costa Rica.

## Author Contributions

**Conceptualization:** Jan Philipp Bechler, Marco Tschapka.

**Data curation:** Jan Philipp Bechler, Kira Steiner.

**Formal analysis:** Jan Philipp Bechler, Kira Steiner.

**Funding acquisition:** Jan Philipp Bechler, Kira Steiner, Marco Tschapka.

**Investigation:** Jan Philipp Bechler, Marco Tschapka.

**Methodology:** Jan Philipp Bechler, Marco Tschapka.

**Project administration:** Jan Philipp Bechler, Marco Tschapka.

**Resources:** Marco Tschapka.

**Software:** Jan Philipp Bechler, Marco Tschapka.

**Supervision:** Jan Philipp Bechler, Marco Tschapka.

**Visualization:** Jan Philipp Bechler.

**Writing – original draft:** Jan Philipp Bechler.

**Writing – review & editing:** Kira Steiner, Marco Tschapka.

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
