## [Decision Letter · Decision Letter 0]

21 Feb 2022

PONE-D-21-39791Feeding performance of nectar-feeding bats (Phyllostomidae: Glossophaginae) indicates that familiarity with flowers shapes resource utilizationPLOS ONE

Dear Dr. Bechler,

Thank you for submitting your manuscript to PLOS ONE. After careful consideration, we feel that it has merit but does not fully meet PLOS ONE’s publication criteria as it currently stands. Therefore, we invite you to submit a revised version of the manuscript that addresses the points raised during the review process.

Both reviewers are largely positive about the manuscript but each have a number of concerns to improve its clarify and rigor. Both reviewers mention a need for greater clarity in articulating hypotheses and predictions in the introduction. Lines 117-128 come close, and so some modification in wording should help here, particularly L124-128. Specific statements about sample size must be included. Reviewer 2 also commented on the need to include a LMM instead of LM; from my reading of L234-249, it does appear LMMs were used here, given the mention of fixed and random effects, so the authors should be sure to describe these analyses as “linear mixed effects models” rather than “linear regression models” to be as specific as possible and avoid confusion. Lastly, I would also suggest the authors include confidence intervals (Figure 1) and raw data (rather than only means and confidence intervals in Figure 5) in their visualizations.==============================

We look forward to receiving your revised manuscript.

Kind regards,

Daniel Becker

Academic Editor

PLOS ONE

Journal Requirements:

2. Please include a complete copy of PLOS’ questionnaire on inclusivity in global research in your revised manuscript. Our policy for research in this area aims to improve transparency in the reporting of research performed outside of researchers’ own country or community. The policy applies to researchers who have travelled to a different country to conduct research, research with Indigenous populations or their lands, and research on cultural artefacts. The questionnaire can also be requested at the journal’s discretion for any other submissions, even if these conditions are not met.  Please find more information on the policy and a link to download a blank copy of the questionnaire here: https://journals.plos.org/plosone/s/best-practices-in-research-reporting. Please upload a completed version of your questionnaire as Supporting Information when you resubmit your manuscript.”

Reviewers' comments:

Reviewer's Responses to Questions

**Comments to the Author**

1. Is the manuscript technically sound, and do the data support the conclusions?

Reviewer #1: Yes

Reviewer #2: Partly

2. Has the statistical analysis been performed appropriately and rigorously? 

Reviewer #1: Yes

Reviewer #2: No

3. Have the authors made all data underlying the findings in their manuscript fully available?

Reviewer #1: Yes

Reviewer #2: Yes

4. Is the manuscript presented in an intelligible fashion and written in standard English?

Reviewer #1: Yes

Reviewer #2: Yes

5. Review Comments to the Author

Reviewer #1: This is an interesting and timely paper. The comparison of two species of nectar-feeding bats is thorough and thoughtful. I think that the manuscript could be revised to increase its impact.

First, the authors should clearly articulate the hypothesis that guided the work. Then they should identify specific predictions they can test with their data. While the authors cruise close to this approach, they could be more specific and clearly distinguish between hypothesis and prediction.

I admire the authors' approach and candor concerning which of their expectations were supported (or not supported) by the data). Often things did not emerge as expected. What does this mean? It could mean that the predictions are not sharp enough, perhaps suggesting that some underlying aspect of the system they use is not correctly presented. Alternatively the bats may be generalist enough to behave in unexpected ways.

Evidence from gut biome and isotope research reveals that classifications of bats by diet are misleading at best, and perhaps confusing as well. There also is evidence that at least Glossophaga soricina relies much more on insects as food than suggested by their designation as nectar-feeders.

I think that the data from gut biomes and isotopes are relevant to understanding the current situation.

In short, this is a neat paper. The authors have done an excellent job, especially in getting others (at least me) to think about the situation in different ways.

Brock Fenton

Reviewer #2: Comments to the Author

I revised the ms entitled “Feeding efficiency of nectar-feeding bats (Phyllostomidae: Glossophaginae) indicates that familiarity with flowers shapes resource utilization”. The study explores the drinking efficiency of two bat species (specialist and generalist) in relation to flower orientation (horizontal and downward-hanging) using real and artificial flowers. The authors quantified time and duration of bat’s visit in each flower and extracted nectar volume per visit. I believe the study makes an important contribution to our understanding of the factors driving bats food selection in field, but I believe the manuscript needs some more work before it is ready for publication. I have four major areas that I encourage the authors to consider:

Mayor points

1 – The first regards to the more precise use of the literature and the lack of hypothesis that do not help understanding the data analysis and results. It is not clear why authors are comparing and focusing on the explanation of many other points if their hypothesis should be to explore the drinking efficiency (volume of extracted nectar) in relation to flower orientation. If you could delimitate and focus on a single research question and their associated predictions it would be easy to understand and interpret the results. I suggest authors to follow the guidelines presented here to help in their revision:

https://conservationbytes.com/2012/10/22/how-to-write-a-scientific-paper/ .

https://www.springernature.com/br/authors/campaigns/writing-a-manuscript/titles-abstracts-keywords

https://journals.plos.org/ploscompbiol/article?id=10.1371/journal.pcbi.1005619

https://www.nature.com/articles/d41586-018-02404-4

2 – The second major point regard to methods, it is not clear for me the total number of each bat species (n=) you used in each experiment, you need to describe the experiments and data collection in more, much more, detail. I added detailed comments below.

3 – The third regards to the statistical analysis, it is not clear if the authors performed a linear regression or a fixed effects model. I actually recommend a GLMM (generalized linear mixed model, or a Linear Mixed Model, LMM) to estimate the effect of bat´s feeding specialization (specialist, generalist) on drinking efficiency (extracted nectar volume) and the different variables of interest (bat size, visit duration, flower species, flower source -natural, artificial-, flower size, flower orientation and flower arrangement) of the single bat individuals (n=?). There are many good sites in the net explaining the use of GLMMs, for example:

https://www.researchgate.net/publication/221995574_Generalized_Linear_Mixed_Models_A_Practical_Guide_for_Ecology_and_Evolution

https://www.frontiersin.org/subjects/generalized-linear-mixed-model

4 – Finally, I think what you have collected is really interesting, but you need to eliminate the lack of clarity for the reader to follow your results and discussion sections by delimitating your research question. Detailed comments below.

Detail feedback

Title

I suggest to change the title to make it clearer:

Bats nectar extraction efficiency of the specialist Hylonycteris underwoodi and the generalist Glossophaga commissarisi (Phyllostomidae:Glossophaginae) in relation to flower orientation

Abstract

L-(29-34): extensive background and did not include de gap.

L-42: the most relevant results were not included.

L-(42-45): it is not clear what you relied on to stand this conclusion.

Introduction

In general, it is better to give information from general to specific topics, from bat´s food selection in field in general, and then to the generalists and specialists nectar feeding bats. Regarding the use of natural nectar feeding sources, and differences within specialist vs generalist bat species there is important literature missing to be included.

L-72: important baselines. However,

L-(82-88): You don`t talk about previous studies that address factors that bats select or not in food, neither in the field, nor in experimental conditions. Nor do you mention previous studies comparing whether specialist bats select the same as generalists or not, and why. Explain it here and then directly connect them with your aims.

Methods

I suggest to include studied species information in this section and to give more specific details about plant species and bat species as the total number of individuals used in experiments, and other relevant information about bats’ ecology and experiments. Also, I suggest to only include data regarding to natural flowers experiments to simplify the main message of the paper.

L-(136-145): These lines do not correspond to Field Site description. Add another subtitle regarding to Bats Collection.

L-(same lines): Did bats were collected the same day of the experiments or did they were maintained in captivity previous days before the experiments? How many? In which conditions? How did you collected bats?

L-(149-151): Does 16% of nectar total concentration corresponds to the natural nectar mean concentration of both plant species? Why did you choose to use this nectar concentration and sugar composition?

L-(same lines): Did you use the same flower species in the same arrangement? Why don’t you changed in another experiment the arrangement among plant species to really test the effect of flower species and/or spatial flower arrangement? How did you control for position effect?

L-188: Why did you choose to use glucose and fructose and not sucrose? You don’t talk about natural nectar sugar composition and total concentration of plant species.

L-197: Why is it necessary to compare an artificial design with a natural stage? What do you expect to obtain? Why don´t you only include results obtained with the most natural field conditions?

L-(216-220) If you are testing bat´s feeding efficiency why it is important to include pollen load in this paper?

Results and Discussion

These sections are long and difficult to follow. Focus only on answering your research question and discussing your main results.

Figures

No figure legends are provided.

6. PLOS authors have the option to publish the peer review history of their article (what does this mean?). If published, this will include your full peer review and any attached files.

Reviewer #1: **Yes: **Melville Brockett Fenton

Reviewer #2: No

---

## [Author Response · Author response to Decision Letter 0]

21 Jul 2022

Academic Editor

Greater clarity in articulating hypotheses and predictions in the introduction. Lines 117-128 come close, and so some modification in wording should help here, particularly L124-128. 

We modified the hypothesis and the predictions and hope that this resulted in better clarity (lines 131-149).

Specific statements about sample size must be included. 

We admit that in some parts the sample size was not clearly stated. We added sample size where missing. 

Reviewer 2 also commented on the need to include a LMM instead of LM; from my reading of L234-249, it does appear LMMs were used here, given the mention of fixed and random effects, so the authors should be sure to describe these analyses as “linear mixed effects models” rather than “linear regression models” to be as specific as possible and avoid confusion. 

Indeed, we used the more general term “linear regression” but calculated in fact a LMM. In response to reviewer 2 we eliminated this expression from the method description to avoid confusion about the used model type and stated that we used “linear mixed effect models from the lme4-package” (line 266). 

Lastly, I would also suggest the authors include confidence intervals (Figure 1) and raw data (rather than only means and confidence intervals in Figure 5) in their visualizations. 

We assume that confidence intervals were suggested for Figure 3 (and not for Figure 1). Confidence intervals were added. For Figure 5 we changed the plot type from mean +/- SE to boxplots to better represent the distribution of raw data. 

The manuscript was revised and edited according to the PLOS ONE’s style requirements. 

The title, authors and affiliations now meet the style requirements. We changed the e-mail address of the corresponding author.

2. Please include a complete copy of PLOS’ questionnaire on inclusivity in global research in your revised manuscript. Our policy for research in this area aims to improve transparency in the reporting of research performed outside of researchers’ own country or community. The policy applies to researchers who have travelled to a different country to conduct research, research with Indigenous populations or their lands, and research on cultural artefacts. The questionnaire can also be requested at the journal’s discretion for any other submissions, even if these conditions are not met. Please find more information on the policy and a link to download a blank copy of the questionnaire here: https://journals.plos.org/plosone/s/best-practices-in-research-reporting. Please upload a completed version of your questionnaire as Supporting Information when you resubmit your manuscript.”

A copy of PLOS’ questionnaire on inclusivity in global research is included and attached as Supporting Information. 

We included a full ethics statement into the ‘Methods’ section containing the information of the local approving authority and the respective permit numbers. 

Captions for the Supporting Information files were attached at the end of our manuscript and we updated the in-text citations. 

The reference list was reviewed to ensure that it is complete and correct. 

Reviewer #1 (Brock Fenton)

First, the authors should clearly articulate the hypothesis that guided the work. Then they should identify specific predictions they can test with their data. While the authors cruise close to this approach, they could be more specific and clearly distinguish between hypothesis and prediction.

Thank you for your observation. We addressed the issue and reviewed the hypotheses and predictions to increase clarity.

I admire the authors' approach and candor concerning which of their expectations were supported (or not supported) by the data). Often things did not emerge as expected. What does this mean? It could mean that the predictions are not sharp enough, perhaps suggesting that some underlying aspect of the system they use is not correctly presented. Alternatively the bats may be generalist enough to behave in unexpected ways.

Evidence from gut biome and isotope research reveals that classifications of bats by diet are misleading at best, and perhaps confusing as well. There also is evidence that at least Glossophaga soricina relies much more on insects as food than suggested by their designation as nectar-feeders. 

I think that the data from gut biomes and isotopes are relevant to understanding the current situation. 

We intentionally stuck with our initial hypotheses with which we started the project and did not just adjust our hypothesis to our findings, although we admittedly learned a lot about our study species in the progress of the project (which is ultimately the purpose of science ;-) ). It is true that the Glossophaga species are considered as generalist, however we are convinced that the generalistic feeding mode is not an inherit trait of this species, but rather a strategy to cope with local or temporal nectar scarcity. These bats show distinct morphological and physiological adaptations to nectarivory and always prefer nectar when it is sufficiently available and only supplement their diet with other sources (insects, fruits) when nectar is scarce. For the area of La Selva, this is nicely demonstrated by seeds of Piper auritum in the feces of G. commissarisi (Tschapka, 2005). These infructescences are available all year-round but seeds are only evident in feces of G. commissarisi during the time of low flower abundance. Thus we are confident that the classification of G. commissarisi as nectarivorous bat species is valid for our study area. It is crucially important to collect year-round data for a specific site to fully understand the meaning of certain food resources for the respective species. Data from only short periods (like in most isotope or microbiome analyses) only provide temporal snapshots and thus are not suitable for making general statements on the degree of specialization of a species. For La Selva, we thankfully have these long-term data and have solid background information about the bats and their resources over the annual cycle (e.g., Tschapka 2004, 2005, Becker et al. 2010, Rothenwöhrer et al 2011). We now also emphasize this argument in our manuscript (lines 109 – 122). 

In short, this is a neat paper. The authors have done an excellent job, especially in getting others (at least me) to think about the situation in different ways.

Thank you!

Reviewer #2 (anonymous)

Major points:

1 – The first regards to the more precise use of the literature and the lack of hypothesis that do not help understanding the data analysis and results. It is not clear why authors are comparing and focusing on the explanation of many other points if their hypothesis should be to explore the drinking efficiency (volume of extracted nectar) in relation to flower orientation. If you could delimitate and focus on a single research question and their associated predictions it would be easy to understand and interpret the results. I suggest authors to follow the guidelines presented here to help in their revision:

https://conservationbytes.com/2012/10/22/how-to-write-a-scientific-paper/ .

https://www.springernature.com/br/authors/campaigns/writing-a-manuscript/titles-abstracts-keywords

https://journals.plos.org/ploscompbiol/article?id=10.1371/journal.pcbi.1005619

https://www.nature.com/articles/d41586-018-02404-4

Thank you for your observation! You helped us to understand where our manuscript might be misleading. Actually, we did not intend to study the role of flower orientation on the feeding performance of nectar-feeding bats. Although we discuss this as one of the main explanations for our findings, it is not the main aim of the study. We happily encourage further research on this topic as we think this would be a great starting point for further studies. So far, only little is known about the role of floral orientation. While it might be a substantial factor driving the interaction, the detailed investigation of floral orientation would have exceeded the scope of this study and thus was included only as a part of the discussion. However, thanks to your comment, we understand that the argument of flower orientation was too present throughout the manuscript, thus giving the impression that this was the main objective of this study. In the revised manuscript we therefore restricted the emphasis on the differential flower orientation between the two focal plant species to the Discussion, to help emphasizing the main purpose of our study. 

2 – The second major point regard to methods, it is not clear for me the total number of each bat species (n=) you used in each experiment, you need to describe the experiments and data collection in more, much more, detail. I added detailed comments below.

Thank you for pointing this out. We admit that the sample sizes were not always clearly stated and we added sample sizes to help clarification where needed. 

3 – The third regards to the statistical analysis, it is not clear if the authors performed a linear regression or a fixed effects model. I actually recommend a GLMM (generalized linear mixed model, or a Linear Mixed Model, LMM) to estimate the effect of bat´s feeding specialization (specialist, generalist) on drinking efficiency (extracted nectar volume) and the different variables of interest (bat size, visit duration, flower species, flower source -natural, artificial-, flower size, flower orientation and flower arrangement) of the single bat individuals (n=?). There are many good sites in the net explaining the use of GLMMs, for example:

https://www.researchgate.net/publication/221995574_Generalized_Linear_Mixed_Models_A_Practical_Guide_for_Ecology_and_Evolution

https://www.frontiersin.org/subjects/generalized-linear-mixed-model

Indeed we used the more general term “linear regression model”, while actually we did specifically an LMM. Based on you comment, we eliminated this expression from the method description to avoid confusion about the used model type and stated clearly that we used “linear mixed effect models from the lme4-package” (lines 266). 

4 – Finally, I think what you have collected is really interesting, but you need to eliminate the lack of clarity for the reader to follow your results and discussion sections by delimitating your research question. Detailed comments below.

Thank you, we really appreciate your helpful comments. We hope that with our revisions we increased the clarity of our manuscript.

Detail feedback

I suggest to change the title to make it clearer:

Bats nectar extraction efficiency of the specialist Hylonycteris underwoodi and the generalist Glossophaga commissarisi (Phyllostomidae:Glossophaginae) in relation to flower orientation.

As explained above, we restricted the emphasis on floral orientation to the discussion as its only one aspect explaining our findings. However, we agree that the title needed some improvement and changed it to “Feeding performance of nectar-feeding bats (Phyllostomidae: Glossophaginae) at flowers of two key-resource plant species”.

Abstract

L-(29-34): extensive background and did not include de gap.

Based on your comment, we now specifically added a remark on the lack of studies on real flowers for this kind of studies (lines 32 - 33).

L-42: the most relevant results were not included.

We now added the most relevant results (lines 45 – 47), which should now help to better understand the following conclusion. 

L-(42-45): it is not clear what you relied on to stand this conclusion.

In combination with the main results we hopefully now clarified the interpretation of the outcome of the study.

Introduction

In general, it is better to give information from general to specific topics, from bat´s food selection in field in general, and then to the generalists and specialists nectar feeding bats. Regarding the use of natural nectar feeding sources, and differences within specialist vs generalist bat species there is important literature missing to be included. 

We added further information on the bat species and their food selection in natural habits to the Introduction to help better understand the framework of our study (lines 93 – 122).

L-72: important baselines. However,

Thank you. We changed our writing accordingly. Splitting this sentence into two definitely increases readability. 

L-(82-88): You don`t talk about previous studies that address factors that bats select or not in food, neither in the field, nor in experimental conditions. Nor do you mention previous studies comparing whether specialist bats select the same as generalists or not, and why. Explain it here and then directly connect them with your aims. 

We hope that our changes now emphasize the actual lack of such studies and provide sufficient (long-term) background information on the general resource situation of nectar-feeding bats in the La Selva area that is necessary to understand the frame of our study and why this is a great and unique place to do exactly this kind of study there. To our very best knowledge, there are only very few studies focusing on nectar-feeding bats of different degrees of specialization. 

Methods

I suggest to include studied species information in this section and to give more specific details about plant species and bat species as the total number of individuals used in experiments, and other relevant information about bats’ ecology and experiments. Also, I suggest to only include data regarding to natural flowers experiments to simplify the main message of the paper.

We added further information on the species into the Introduction and we are confident that the provided information now suffices to understand the ecology of the focal species. 

We adjusted the description of the Methods, especially we emphasized that only one bat at a time was used in the experiments as we understand that it was not clearly stated before (lines 168 – 176). 

You suggested to only including data on real flowers in the paper. Thus far, the majority of studies on the feeding performance of nectar-feeders were conducted with artificial flowers (ideal scenarios) and the obtained insights were then transferred to natural systems. Our study provides now also the opportunity to test the transferability of such approaches as we can directly compare it with the feeding performance at real flowers. As we unexpectedly observed a substantially higher caloric yield at real flowers compared to the artificial setup in H. underwoodi we are convinced that this result is valuable information, and emphasizes the need for more studies with real flowers in order to better understand the complex spectrum of animal

---

## [Decision Letter · Decision Letter 1]

29 Sep 2022

PONE-D-21-39791R1Feeding performance of nectar-feeding bats (Phyllostomidae: Glossophaginae) at flowers of two key-resource plant speciesPLOS ONE

Dear Dr. Bechler,

 Thank you for submitting your manuscript to PLOS ONE. After careful consideration, we feel that it has merit but does not fully meet PLOS ONE’s publication criteria as it currently stands. Therefore, we invite you to submit a revised version of the manuscript that addresses the points raised during the review process.

 Please submit your revised manuscript by Nov 13 2022 11:59PM. If you will need more time than this to complete your revisions, please reply to this message or contact the journal office at plosone@plos.org. Please include the following items when submitting your revised manuscript:A rebuttal letter that responds to each point raised by the academic editor and reviewer(s). You should upload this letter as a separate file labeled 'Response to Reviewers'.A marked-up copy of your manuscript that highlights changes made to the original version. You should upload this as a separate file labeled 'Revised Manuscript with Track Changes'.An unmarked version of your revised paper without tracked changes. You should upload this as a separate file labeled 'Manuscript'.

We look forward to receiving your revised manuscript.

Kind regards,

Daniel Becker

Academic Editor

PLOS ONE

Reviewers' comments:

Reviewer's Responses to Questions

**Comments to the Author**

1. If the authors have adequately addressed your comments raised in a previous round of review and you feel that this manuscript is now acceptable for publication, you may indicate that here to bypass the “Comments to the Author” section, enter your conflict of interest statement in the “Confidential to Editor” section, and submit your "Accept" recommendation.

Reviewer #2: (No Response)

2. Is the manuscript technically sound, and do the data support the conclusions?

Reviewer #2: Partly

3. Has the statistical analysis been performed appropriately and rigorously? 

Reviewer #2: Yes

4. Have the authors made all data underlying the findings in their manuscript fully available?

Reviewer #2: No

5. Is the manuscript presented in an intelligible fashion and written in standard English?

Reviewer #2: No

6. Review Comments to the Author

Reviewer #2: In most of the answers to my previous requests in the last revision, it seems that authors responded and completely attended to them. However, they do not include the modified words, or punctually the requested changes in the answer to reviewers, and sometimes, they did not include the lines, and when I read the whole section I find that changes are not included. Please attend in detail to my concerns and be more specific since many of my previous requests were not met in detail.

As in the previous revision, the aims are not connected with what you are fully explaining in the introduction. You talk a lot about plants and your study is about bat selection, you are not talking about this, and if you are measuring feeding efficiency why don’t you talk about it deeply and explain which traits on nectar and feeding resources bats rely their food selection?

Clarify what is feeding performance.

How did you measure feeding performance? Since you say you calculated nectar extraction efficiency and caloric uptake are them all the same? If not, why are these variables not included in your objectives and hypotheses? Please clarify first concepts in the introduction and here in methods the way you quantified each variable, cause it seems that methods are not connected with the introduction.

Again, for me is not clearly justified here why performing experiments in both, natural and experimental conditions. what is the difference in the information provide by each approach?

Results are confusing and not easy to follow, and I can`t find the most important contribution of the study. Please follow the same order in studied variables along the whole manuscript.

Discussion section is quite long and imprecise, the most important results are not well discussed.

7. PLOS authors have the option to publish the peer review history of their article (what does this mean?). If published, this will include your full peer review and any attached files.

Reviewer #2: **Yes: **Rodríguez-Peña

---

## [Author Response · Author response to Decision Letter 1]

15 Apr 2023

Reviewer #2:

In most of the answers to my previous requests in the last revision, it seems that authors responded and completely attended to them. However, they do not include the modified words, or punctually the requested changes in the answer to reviewers, and sometimes, they did not include the lines, and when I read the whole section I find that changes are not included. Please attend in detail to my concerns and be more specific since many of my previous requests were not met in detail.

As in the previous revision, the aims are not connected with what you are fully explaining in the introduction. You talk a lot about plants and your study is about bat selection, you are not talking about this, and if you are measuring feeding efficiency why don’t you talk about it deeply and explain which traits on nectar and feeding resources bats rely their food selection?

Clarify what is feeding performance.

Thank you for you comment and feedback on our manuscript. To clarify, our study did not specifically target food selection, but rather investigated the role of plants in bats’ foraging behavior and especially their feeding efficiency, but also hovering duration and nectar uptake (summarized as “feeding performance”; see comment below). Besides plants traits we also discuss the role of bats’ familiarity with and dependence on these plants in achieving this feeding performance. Regarding this topics, there are currently few studies addressing this aspects with real instead of artificial flowers. However, with this the present study we hope to contribute to close this gap in knowledge and also encourage further work on this fascinating topic. 

How did you measure feeding performance? Since you say you calculated nectar extraction efficiency and caloric uptake are them all the same? If not, why are these variables not included in your objectives and hypotheses? Please clarify first concepts in the introduction and here in methods the way you quantified each variable, cause it seems that methods are not connected with the introduction.

We apologize for the confusion in our previous response. Let us address your questions and provide more clarity on the issue. We agree that the use of the term “feeding performance” might be confusing to the reader. This expression is intended to include the different variable we addressed in this study, i.e. hovering duration, nectar uptake, feeding efficiency; thus being a more general expression. We specified this in L35 & L36 in the abstract, in L98 & L99 in the introduction and L216 & L217 in the methods.

Again, for me is not clearly justified here why performing experiments in both, natural and experimental conditions. what is the difference in the information provide by each approach?

We further clarified the justification for this aspect of the study (L171 – L173). Collecting data on feeding behavior at real flowers now gives us the unique opportunity to compare the artificial approach (what was usually done in the past) to the natural condition. This provides insight into the validity of the information gained from experimental with artificial flowers.

Results are confusing and not easy to follow, and I can`t find the most important contribution of the study. Please follow the same order in studied variables along the whole manuscript.

We made sure that we followed the same order of variables for both sections (comparison between bats and comparison between plant species) and also maintain the same order in the discussion. However, as we compare both, differences between bat species and differences between plant species, this dual approach requires occasionally deviating from this order, with the intention to keep up the logical flow in the discussion.

Discussion section is quite long and imprecise, the most important results are not well discussed.

Again, thank you for the time you took to review our paper and we appreciate your feedback. We understand you concerns regarding length and precision of the discussion section. However, given the complexity of the results and after careful consideration, we think that the current version provides a thorough and comprehensive discussion of our results, including their significance and implications. We would appreciate if you could provide us with a bit more specific feedback regarding which important results are not well discussed, as this would enable us to address your concerns more specifically.

ABSTRACT

In the abstract section any of the 3 major changes I asked for in the previous revision were attended. They were: extensive background and did not include the gap, the most relevant results were not included and it is not clear what you relied on to stand this conclusion.

L. 29-37 Very large and general background, please be more accurate to your study question.

We shortened and streamlined this part substantially.

L.38 It is not yet clear stated how did you measure feeding performance.

L35 & L36: Feeding performance, as a general term for the measured aspects (nectar uptake, hovering duration and the resulting feeding efficiency) is now stated in the abstract.

L 40-45 Please rewrite these sentences since you included mixed sections, hypothesis and methods, making the abstract very difficult to catch the most important message. It is not clear what and how you measure, what and how you compare, between plant species? Between two bat species? Between selection of the specialist and generalist bats?

L38 & L39: Thank you for your helpful comment. We further specified what we measured (see comment above) as well as what we compared. We also aimed to make the difference between methods and hypotheses more clear. 

L.42 Bromeliad?

Werauhia gladioliflora is a member of the Bromeliaceae family and here we aimed to provide brief systematic information on the studies plant species to the interested reader.

L.45 The results section did not match to what I would expect reading the previous lines, as I would expect to see results for 2 bat species interacting with two plant species, and then the comparison between specialist vs generalist bats feeding performance. Instead, you only included to present results for H. underwoodi at only one plant species (M. neuranthum). If no differences with the generalist bat species and the other plants were found, please state it clearly here.

L43-L49: We revised the commented part of the abstract and now clearly mention the results on both bats species and on both data from real flowers as well as results from the comparison between real and artificial flowers. We hope we now provide a satisfactory level of information.

L. 47 How did you arrive to say this?

Please see comment above.

L.50 You are not providing a general closing with your most important finding.

L68-L72: A general closing is now provided in at the end of the section.

INTRODUCTION

L. 138 In which previous studies are you relying your hypotheses? Further, in the previous paragraphs you do not talk about on which you included in your hypotheses.M

We agree that it is important to provide a clear understanding of the existing literature and how our work fits into it. However, our study investigates a novel topic that has not been previously explored in the literature. Therefore, we were not able to refer to any previous studies in developing our hypothesis, but simply base this assumption on the nectar-rich nature of the Werauhia flowers as it is reasonable to assume that larger nectar-volumes in a flower facilitate a higher feeding efficiency. We now specify this notion in this part, hoping to have clarified our hypothesis. We have taken your comment into consideration and clarify how our hypotheses were developed.

MATERIAL AND METHODS

L.227 experiments

Thank you for your correction. We changed experiment to the plural form.

---

## [Decision Letter · Decision Letter 2]

27 Jun 2023

PONE-D-21-39791R2Feeding performance of nectar-feeding bats (Phyllostomidae: Glossophaginae) at flowers of two key-resource plant speciesPLOS ONE

Dear Dr. Bechler,

Thank you for submitting your manuscript to PLOS ONE. After careful consideration, we feel that it has merit but does not fully meet PLOS ONE’s publication criteria as it currently stands. Therefore, we invite you to submit a revised version of the manuscript that addresses the points raised during the review process.

We look forward to receiving your revised manuscript.

Kind regards,

Daniel Becker

Academic Editor

PLOS ONE

Reviewers' comments:

Reviewer's Responses to Questions

**Comments to the Author**

1. If the authors have adequately addressed your comments raised in a previous round of review and you feel that this manuscript is now acceptable for publication, you may indicate that here to bypass the “Comments to the Author” section, enter your conflict of interest statement in the “Confidential to Editor” section, and submit your "Accept" recommendation.

Reviewer #1: (No Response)

Reviewer #2: (No Response)

2. Is the manuscript technically sound, and do the data support the conclusions?

Reviewer #1: Yes

Reviewer #2: Partly

3. Has the statistical analysis been performed appropriately and rigorously? 

Reviewer #1: N/A

Reviewer #2: Yes

4. Have the authors made all data underlying the findings in their manuscript fully available?

Reviewer #1: Yes

Reviewer #2: No

5. Is the manuscript presented in an intelligible fashion and written in standard English?

Reviewer #1: Yes

Reviewer #2: No

6. Review Comments to the Author

Reviewer #1: This manuscript is overwhelming because it presents a great deal of information without a clear context.

What is the hypothesis that underlies the work? What specific (numbered) predictions arising from the hypothesis that the authors can test with the data. Adding this information will dramatically increase the impact of the work by making it clearer to the readers, whether expert of interested colleague.

Tighten the writing - make clear statements. One subject per sentence, subject at the beginning of the sentence. Make sure of the spelling and word choice.

I think that this manuscript has the potential to be a classic. But both previous reviewers have suggested changes that have not clearly been used to advantage by the authors. Make sure that you need to present all of the details ... consider omitting tangential material to improve the focus of the manuscript.

Brock Fenton

Reviewer #2: This is an interesting study that contributes to the knowledge of the use of feeding resources by nectarivore bats. However, I suggest major changes related to its structure, ideas order, and wording.

Introduction

You talk about optimal foraging theory, implications of foraging behaviour, pollinator-plant interactions, plants rewards, nectar uptake, however, it is not clearly defined and delimited what is feeding performance. This is the main concept you should develop in the introduction, which variables are included in this concept, which of them are already studied in natural and artificial conditions and which are not, considering plant species and pollinator types.

What do you expect to see in natural conditions and what in artificial conditions, what are both approaches important to include?

The objective stated the comparison of drinking efficiency, is it ok? What about the other variables in “feeding performance”?; if so, then this must be specified in the tittle, it is confusing, or either stay all feeding performance variables you are going to test here. Are the same feeding performance variables going to be tested in both, natural and artificial conditions?

You stated hypotheses, however, you did not included citations on which you rely them.

I suggest that the ideas be organized as follows: First paragraph- Define feeding performance and which variables are considered in this concept; Second paragraph- State the background, in-depth review of the literature, addressing the state of the art of each of the feeding performance variables, under natural and artificial conditions by pollinator type and plants. End the paragraph talking about bats background and gaps. Third paragraph- 1) Clearly stay the aim of the study. 2) Clearly justify why you are using natural and artificial assays. Explain here why you measure hovering duration (energetic investment), extracted volume (caloric uprake) and standardized nectar extraction efficiency. 3) Clearly justify why you are using two bat species and two plant species.

Material and methods

193-194 It says “We aimed to quantify the feeding performance (i.e. hovering duration, nectar uptake, and feeding efficiency)” which do not match with what you stated in your objective.

Explain in the same order how did you measure each of: 1) hovering duration, 2) extracted volume and 3) standardized nectar extraction efficiency.

L 258-262 It is not clear why including pollen samples in bats, since you did not say anything about in the introduction, objective and hypotheses. How important is it in feeding performance? If so, you should include it throughout the introduction.

Results

Present results as:

Natural flowers

1) Energetic investment, 2) Caloric uptake and 3) standardized nectar extraction efficiency.

Artificial flowers

2) Energetic investment, 2) Caloric uptake and 3) standardized nectar extraction efficiency.

In the case tested variables are the same in both, natural and artificial flowers. Is confussing since in artificial flowers it says “standardized nectar uptake” “standardized caloric yield” “caloric net yield”. It is not clear at all if tested variables are the same in natural and artificial flowers, is o, please use the same terms throughout the manuscript. If not please clearly explain in the last paragraph of the introduction as suggested and in the methods section.

Discussion

Discussion is a confusing and quite long section. Please present the discussion section in the same order than in results. Stay the most important findings one by one. For each one, explain your finding, physiological and ecological implications, then compare your finding with previous literature.

Conclusion

Only refer to highlight your most important findings.

7. PLOS authors have the option to publish the peer review history of their article (what does this mean?). If published, this will include your full peer review and any attached files.

Reviewer #1: **Yes: **Melville Brockett Fenton

Reviewer #2: No

---

## [Author Response · Author response to Decision Letter 2]

5 Nov 2023

4. Have the authors made all data underlying the findings in their manuscript fully available?

Reviewer #1: Yes

Reviewer #2: No

Response JPB: All the necessary data have indeed been uploaded as part of the supplementary materials accompanying our submission (see S1_Data). Reviewer #1 has also acknowledged that all data has been provided. If there are still any specific concerns please do not hesitate to let us know. We are committed to providing all the necessary information to support a thorough evaluation of our work.

5. Is the manuscript presented in an intelligible fashion and written in standard English?

Reviewer #1: Yes

Reviewer #2: No

Response JPB: In regard to the feedback concerning the quality of the English language in our publication, we understand the importance of adhering to language standards in scientific communication. We have once more tried to shorten and improve the languange and hope that it now meets the approval of both referees.

For both reviewers: Thank you very much for your valuable feedback on our paper. We followed the recommendations of the referees and shortened the manuscript substantially. We decided to focus on the behavior of the bats when feeding on the two natural flowers and removed the part about artificial flowers as we understand that presenting both parts caused some confusion and resulted in an overwhelming paper. Further, we also removed the short section about pollen that was merely intended to demonstrate the high usage of Merinthopodium by Hylonycteris throughout the year and the resulting high familiarity with these flowers, instead we refer to former papers. We also removed the apparently misleading term “feeding performance” throughout the paper and just refer to feeding efficiency, which is clearly defined in the Methods section. We adapted the rest of the paper to these changes and hopefully also cleaned up the language. We sincerely hope that this makes the paper more straightforward and easier to understand.

---

## [Decision Letter · Decision Letter 3]

3 Jan 2024

PONE-D-21-39791R3Feeding efficiency of two coexisting nectarivorous bat species (Phyllostomidae: Glossophaginae) at flowers of two key-resource plantsPLOS ONE

Dear Dr. Bechler,

Thank you for submitting your manuscript to PLOS ONE. After careful consideration, we feel that it has merit but does not fully meet PLOS ONE’s publication criteria as it currently stands. Therefore, we invite you to submit a revised version of the manuscript that addresses the points raised during the review process.

We look forward to receiving your revised manuscript.

Kind regards,

Daniel Becker

Academic Editor

PLOS ONE

Journal Requirements:

Reviewers' comments:

Reviewer's Responses to Questions

**Comments to the Author**

1. If the authors have adequately addressed your comments raised in a previous round of review and you feel that this manuscript is now acceptable for publication, you may indicate that here to bypass the “Comments to the Author” section, enter your conflict of interest statement in the “Confidential to Editor” section, and submit your "Accept" recommendation.

Reviewer #1: (No Response)

2. Is the manuscript technically sound, and do the data support the conclusions?

Reviewer #1: (No Response)

3. Has the statistical analysis been performed appropriately and rigorously? 

Reviewer #1: (No Response)

4. Have the authors made all data underlying the findings in their manuscript fully available?

Reviewer #1: (No Response)

5. Is the manuscript presented in an intelligible fashion and written in standard English?

Reviewer #1: (No Response)

6. Review Comments to the Author

Reviewer #1: For the most part the authors have addressed the points raised by earlier reviewers. I think that there is one additional change that they should consider. Clearly state the hypothesis that underlies the work ... this has been done. But now identify specific numbered predictions that arise from the hypothesis, and that they can test with their data. Not having the specific predictions camouflages the importance and focus of the research. Specific predictions can guide the reader through the manuscript, increasing the impact of the work.

Neat study, just add a small finishing touch

Brock Fenton

7. PLOS authors have the option to publish the peer review history of their article (what does this mean?). If published, this will include your full peer review and any attached files.

Reviewer #1: **Yes: **Brock Fenton

---

## [Author Response · Author response to Decision Letter 3]

31 Mar 2024

Thank you for the opportunity to revise our manuscript titled " Feeding efficiency of two coexisting nectarivorous bat species (Phyllostomidae: Glossophaginae) at flowers of two key-resource plants." We appreciate the time and effort that the reviewers have dedicated to evaluating our work, and we are grateful for their constructive feedback.

We are pleased to hear that the reviewers found our study to be generally well-executed and insightful. However, we acknowledge the concerns raised regarding the clarity of our hypothesis. Upon careful review, we have made adjustments to ensure that the hypotheses are clearly articulated and made numbered and specific predictions.

We are confident that this revision will address the reviewers' concerns and enhance the clarity of our manuscript. We appreciate the opportunity to improve our work and look forward to the possibility of its publication in PLOS ONE.

---

## [Editor Report · Decision Letter 4]

23 Apr 2024

Feeding efficiency of two coexisting nectarivorous bat species (Phyllostomidae: Glossophaginae) at flowers of two key-resource plants

PONE-D-21-39791R4

Dear Dr. Bechler,

We’re pleased to inform you that your manuscript has been judged scientifically suitable for publication and will be formally accepted for publication once it meets all outstanding technical requirements.

Kind regards,

Daniel Becker

Academic Editor

PLOS ONE
---

## [Editor Report · Acceptance letter]

18 Jun 2024

PONE-D-21-39791R4 

PLOS ONE

Dear Dr. Bechler, 

I'm pleased to inform you that your manuscript has been deemed suitable for publication in PLOS ONE. Congratulations! Your manuscript is now being handed over to our production team.

Kind regards, 

on behalf of

Daniel Becker 

Academic Editor

PLOS ONE